



# Multipoint Reconstruction of Wind Speeds

Christian Behnken, Matthias Wächter, and Joachim Peinke

Institute of Physics/ForWind, University of Oldenburg.

**Correspondence:** Peinke (peinke@uni-oldenburg.de)

**Abstract.** The most intermittent behavior of atmospheric turbulence is found for very short time scales. Based on a concatenation of conditional probability density functions (cpdfs) of nested wind speeds increments, inspired by a Markov process in scale, we derive a short-time predictor for wind speed fluctuations around a non-stationary mean value and with a corresponding non-stationary variance. As a new quality this short time predictor enables a multipoint reconstruction of wind data. The used cpdfs are (1) directly estimated from historical data from the offshore research platform FINO1 and (2) obtained from numerical solutions of a family of Fokker-Planck equations in the scale domain. The explicit forms of the Fokker-Planck equations are estimated from the given wind data. A good agreement between the statistics of the generated synthetic wind speed fluctuations and the measured is found even on time scales below 1s. This shows that our approach captures the short-time dynamics of real wind speed fluctuations very well. Our method is extended by taking the non-stationarity of the mean wind speed and its non-stationary variance into account.

## 1 Introduction

The transition of our energy system, formerly strongly relying on gas and coal, to a decarbonized one, mainly based on wind and solar power, is still ongoing work, but great progress has been made: From 2005 to 2017 the share in installed capacity of Wind (Solar) has been increased from 6% (0.3%) to 18% (11.5%) in the European Union (WindEurope, 2018). The downside of this increasing share of those fluctuating renewable energy sources is their integration into the power grid. By analyzing measurements of fed-in wind and solar power it could be shown that their fluctuations strongly deviate from Gaussian behavior on time scales ranging from hours to seconds (Anvari et al., 2016) and for wind power even for scales below 1s (Haehne et al., 2018). This survival of the atmospheric intermittency in the power grid faces the grid operators with the great challenge to ensure stable power supply, even under highly volatile conditions.

To aid the design of our future energy systems, for example to size the needed energy storage or the power generation capacity of conventional power plants, much work has been done in the field of long term wind speed/power modeling, utilizing Markov chain models. Whereas simple first-order Markov chain models cannot grasp the characteristics of long term correlations of wind speeds (Brokish and Kirtley, 2009), higher-order Markov chain models perform better, but will require more input data for





estimating the transitions matrices or some simplifications ((Pesch et al., 2015), (Brokish and Kirtley, 2009), (Papaefthymiou and Klockl, 2008), (Weber et al., 2018)).

Despite their dramatic effect of long range correlation and fluctuations of wind speeds on the power generation (and thus the grid stability), wind speed fluctuations are known be most intermittent on short time scales (Boettcher et al., 2003). Models considering time steps ranging from minutes to seconds or even below are of course not suited for energy system analysis on

national levels, but are useful tools for wind turbine operators. For a time step of 10min (Carpinone et al., 2010) presented a higher-order Markov chain model for wind power and (Milan et al., 2013) showed a stochastic power model based on a conditional Langevin equation to work even in the range of seconds.

The knowledge of full three-dimensional wind fields for all three velocity components and the pressure in all details would be desirable. The missing of the basic understanding, the impracticability of handling such huge data sets as well as the complexity

of the wind energy conversion process leads often to the demand of simplified models for wind speed. Common approaches for the design of wind turbines are the so-called Mann uniform shear and the Kaimal spectral and exponential coherence model (iec, 2005). Both models take spectral and coherence aspects of turbulent velocity fluctuations in account, thus handling the fluctuations as Gaussian distributed and stationary. Higher order statistical effects like the prominent intermittency effect of turbulence and non-stationarities are not taken into account, see for example (Mücke et al., 2011). Another approach is to use

one-dimensional effective wind speed time series, representing for example the wind field together with the rotor aerodynamics as it impacts the drive train or can be used to model the above discussed energy conversion process (Wächter et al., 2011).

Within this work we propose a novel stochastic generator of one-dimensional wind speed fluctuations with a sampling interval of 0.1s. One main novelty is that we show how to grasp by this model multipoint statistics of wind structures in time. Such a stochastic multipoint approach should in principle be able to grasp wind structures like gust as well as clustering of

wind fluctuations. The method was initially developed by (Nawroth and Peinke, 2006) in the context of homogeneous isotropic turbulence and later on applied to the modeling of log-return rates of current exchange rates (Nawroth et al., 2010) and and velocity increments of idealised homogeneous isotropic turbulence (Stresing and Peinke, 2010). The successfull application to ocean gravity waves (Hadjihosseini et al., 2016) showed that structures of monster waves can be grasped by this approach correctly (Hadjihoseini et al., 2018). For a recent review see (Peinke et al., 2019). Finally we want to point out that we show

also how to handle the aspect non-stationary wind conditions.

We will continue as follows: In a first part we discuss the method for a subset of wind data characterised by its mean wind speed and its standard deviation. For such data it is shown how, arising from a Langevin process in a scale, a predictor for the upcoming wind speed fluctuation around a mean value can be derived by a nesting of conditional probability density functions. Afterwards we check for Markovian properties of the wind speed fluctuations in scale and set up a Fokker-Planck equation,

corresponding to the Langevin process in scale, and show how it contributes to the improvement of our stochastic prediction method. Finally in the second part the non-stationary mean wind speed and its non-stationary variance is incorporated into our approach to achieve more realistic wind speed time series.



## 2  Method

In this section we present the stochastic framework used for our multipoint reconstruction scheme. As a simplification we start

this discussion for 1 min blocks of wind data $U(t)$ which share the same mean wind speed $\overline{U}$ and the same standard deviation $\sigma_U$, as suggested in (Morales et al., 2012). Fixing $\overline{U}$ and $\sigma_U$ we generate quasi stationary subsets of data. As data we use measured data from the offshore research platform FINO 1: The data were recorded between calender week 1 to 10 in 2007 by an ultrasonic anemometer, mounted at 80m height. The use of the methods for non-stationary time series is outlined in Section 3. Concerning the notations, we abbreviate in this Section stochastic notations like $p_{\overline{U},\sigma_U}(...)$ by $p(....)$, i.e. the discussion is

implicitly restricted to these block condition. The wind speed of such a bock sequence is labeled by $u(t)$, here $u$ is normalised to zero mean and standard deviation of 1.

### 2.1  Multipoint Statistics

Since we assume wind speeds to emerge from a turbulent cascade, increments will take a key role for our method. Having a time series of wind speeds $u(t)$, the corresponding increment time series $\Delta u(\tau)$, depending on a certain scale $\tau$, is given by

$$\Delta u(\tau) = u(t) - u(t - \tau). \tag{1}$$

This is the definition of so called right-sided increments. Note that the calculation of $\Delta u(\tau)$ after this definition depends on the current wind speed $u(t)$ and a past value $u(t-\tau)$, whereas the use of left-sided increments $\Delta u(\tau) = u(t+\tau) - u(t)$ would premise the knowledge of future values $u(t+\tau)$, creating a contradiction as we aim at producing synthetic wind speed time series. To ease readability we use the shorthand notations $u_i := u(t - \tau_i)$ and $\Delta u_i := u(t) - u(t - \tau_i)$ with $\tau_i = i \cdot \tau$ $(i = 1, 2, 3, ...)$

in the following.

As a further remark we note that although we consider in this work time series of wind speed, we often talk of multipoint statistics. $\Delta u(\tau)$ is considered as a statistical two-point quantity, which more correctly could be denoted as two-time quantity. Based on the commonly used hypotheses on frozen turbulence by Taylor, for short time fluctuations time and space statistics are related linearly by the mean wind speed (see also (Peinke et al., 2019)).

Our idea is to predict a wind speed $u^*(t^*)$ only by knowledge of its $N$ past values $\{u_1(t^* - \tau_1), ..., u_N(t^* - \tau_N)\}$. The probability of an event $u^*$ to happen at time $t^*$ can then be expressed by the conditional probability density function (conditional *pdf*)

$$p(u^*, t^* | u_1, t^* - \tau_1; ...; u_N, t^* - \tau_N) =$$
$$\frac{p(u^*, t^*; u_1, t^* - \tau_1; ...; u_N, t^* - \tau_N)}{p(u_1, t^* - \tau_1; ...; u_N, t^* - \tau_N)}, \tag{2}$$

using the definition of conditional probabilities. Note this conditional *pdf* is the key quantity to estimate a new wind speed

value $u^*$ and it can be used iteratively to generate new time series, as we show below.

To link equation (2) to the idea of an underlying turbulent cascade we identify the conditional *pdf* on the lhs with the conditional





*pdf* $p(u^*, t^*|\Delta u_1, t^* - \tau_1; ...; \Delta u_N, t^* - \tau_N)$. Thus the numerator of the rhs of (2) can be rewritten as

$$
\begin{aligned}
p(u^*, t^*; u_1, t^* - \tau_1; ...; u_N, t^* - \tau_N) = \\
p(u^*, u^* - u_1, \tau_1; ...; u^* - u_N, \tau_N)
\end{aligned}
\tag{3}
$$

and the nominator as

$$
\begin{aligned}
p(u_1, t^* - \tau_1; ...; u_N, t^* - \tau_N) = \\
p(u_1; u_1 - u_2, \tau_2 - \tau_1; ...; u_1 - u_N, \tau_N - \tau_1)
\end{aligned}
\tag{4}
$$

The identity of the expressions in (3) and (4) can mathematically rigorously be shown, as done in (Nawroth et al., 2010), but intuitively speaking the sequences on the lhs and rhs must yield the same joint pdf, since the increments on the rhs respectively have a common reference point $u^*$ or $u_1$. Next we factorize the joint pdfs from equations (3) and (4) by iteratively using conditional *pdf*s

$$
\begin{aligned}
p(u^*; \Delta u_1, \tau_1; ...; \Delta u_N, \tau_N) = \\
p(\Delta u_1, \tau_1 | \Delta u_2, \tau_2; ...; \Delta u_N, \tau_N; u^*) \cdot \\
p(\Delta u_2, \tau_2 | \Delta u_3, \tau_3; ...; \Delta u_N, \tau_N; u^*) \cdots \\
p(\Delta u_{N-1}, \tau_{N-1} | \Delta u_N, \tau_N; u^*) \cdot \\
p(\Delta u_N, \tau_N | u^*) \cdot p(u^*)
\end{aligned}
\tag{5}
$$

and with $\tilde{\Delta u_i} := u(t^* - \tau_1) - u(t^* - \tau_i)$ with the time scale $\tau_i - \tau_1$:

$$
\begin{aligned}
p(u_1; \tilde{\Delta u_2}, \tau_2 - \tau_1; ...; \tilde{\Delta u_N}, \tau_N - \tau_1) = \\
p(\tilde{\Delta u_2}, \tau_2 - \tau_1 | \tilde{\Delta u_3}, \tau_3 - \tau_1; ...; \tilde{\Delta u_N}, \tau_N - \tau_1; u_1) \cdot \\
p(\tilde{\Delta u_3}, \tau_3 - \tau_1 | \tilde{\Delta u_4}, \tau_4 - \tau_1; ...; \tilde{\Delta u_N}, \tau_N - \tau_1; u_1) \cdots \\
p(\tilde{\Delta u_{N_1}}, \tau_{N-1} - \tau_1 | \tilde{\Delta u_N}, \tau_N - \tau_1; u_1) \cdot \\
p(\tilde{\Delta u_N}, \tau_N - \tau_1 | u_1) \cdot p(u_1).
\end{aligned}
\tag{6}
$$

A further step in reducing the dimensionality of the involved pdfs can be performed upon assuming the scale process to be Markovian, i.e. there exists a time scale separation $\Delta \tau_{ME} = \tau_j - \tau_i \ (j > i)$, where

$$
\begin{aligned}
p(\Delta u_i, \tau_i | \Delta u_j, \tau_i + \Delta \tau_{ME}; ...; \Delta u_n, \tau_i + n \cdot \Delta \tau_{ME}; u^*) = \\
p(\Delta u_i, \tau_i | \Delta u_j, \tau_i + \Delta \tau_{ME}; u^*)
\end{aligned}
\tag{7}
$$

holds. The time scale separation $\Delta \tau_{ME}$ is often called Markov-Einstein length (Einstein, 1905) and for various systems its existence could be shown empirically, ranging from jet streams in laboratory experiments (Renner et al., 2001), (N Reinke) to large geophysical systems such as ocean gravity waves (Hadjihosseini et al., 2016).



It is known that the evolution of conditional pdfs of a Markov process can be described by the famous Kramers-Moyal

expansion Risken (1996), which notes for our stochastic process in scale $\Delta u_i$

$$
-\tau_i \frac{\partial}{\partial \tau_i} p(\Delta u_i | \Delta u_j; u^*) =
\sum_{n=1}^{\infty} \left( -\frac{\partial}{\partial \Delta u_i} \right)^n \left[ D^{(n)}(\Delta u_i, \tau_i, u^*) p(\Delta u_i | \Delta u_j; u^*) \right], \tag{8}
$$

requiring $\tau_j - \tau_i \geq \Delta \tau_{ME}$ and with the Kramers-Moyal coefficients $D^{(n)}$ being defined as

$$
D^{(n)}(\Delta u_i, \tau_i, u^*) =
\lim_{\Delta \tau \to 0} \frac{\tau_i}{n! \Delta \tau} \langle [\Delta u_i'(\tau_i - \Delta \tau, u^*) - \Delta u_i(\tau_i, u^*)]^n \rangle. \tag{9}
$$

In contrast to the Kramers-Moyal expansion in time domain, a minus sign on the lhs of equation (8) has to be added for scale

processes, since during evolution of the process the scale $\tau$ is decreasing. According to the Pawula theorem the Kramers-Moyal

coefficients vanish for $n \geq 3$, if $D^{(4)} = 0$, thus the Kramers-Moyal expansion reduces to the Fokker-Planck equation (FPE)

$$
-\tau_i \frac{\partial}{\partial \tau_i} p(\Delta u_i | \Delta u_j; u^*) =
-\frac{\partial}{\partial \Delta u_i} \left[ D^{(1)}(\Delta u_i, \tau_i, u^*) p(\Delta u_i | \Delta u_j; u^*) \right] +
-\frac{\partial^2}{\partial \Delta u_i^2} \left[ D^{(2)}(\Delta u_i, \tau_i, u^*) p(\Delta u_i | \Delta u_j; u^*) \right], \tag{10}
$$

with the drift function $D^{(1)}$, accounting for the deterministic evolution of the stochastic process, whereas the so called diffusion

function $D^{(2)}$ scales the amplitude of the noise term of the corresponding Langevin equation

$$
-\frac{\partial}{\partial \tau} \Delta u(\tau, u^*) =
\frac{1}{\tau} D^{(1)}(\Delta u, \tau, u^*) + \sqrt{\frac{1}{\tau} D^{(2)}(\Delta u, \tau, u^*)} \cdot \Gamma(\tau) \tag{11}
$$


with the Gaussian noise $\Gamma(\tau)$, fulfilling $\langle \Gamma(\tau) \rangle = 0$ and as well $\langle \Gamma(\tau) \Gamma(\tau') \rangle = 2\delta_{\tau\tau'}$. This equation directly describes the

evolution of a single trajectory along the scale $\tau$.

As it can be easily seen, the FPE can be used to determine the factorized pdfs from eq. (5) and eq. (6) if the process is Markovian

and higher order Kramers-Moyal coefficients are zero. By inserting those expressions into eq. (2), we finally get

$$
p(u^*, t^* | u_1, t^* - \tau_1; ...; u_N, t^* - \tau_N) =
\frac{p(u^*)}{p(u_1)} \cdot \frac{\Pi_{i=1}^{N-1} p(\Delta u_i | \Delta u_{i+1}; u^*)}{\Pi_{i=2}^{N-1} p(\tilde{\Delta u}_i | \tilde{\Delta u}_{i+1}; u_1)} \cdot \frac{p(\Delta u_N | u^*)}{p(\tilde{\Delta u}_N | u_1)}. \tag{12}
$$


This equivalence between a conditional pdf with $N$ conditions and a nested chain of several conditional pdfs with only two

conditions, stemming from the three-point closure of a cascade process, is extremely helpful if one aims at estimating the pdfs

from measurements, since the high dimensional pdfs in eq. (2) would require a tremendous amount of realizations in order to

be estimated well. For a more detailed discussion of this multipoint approach we refer to (Peinke et al., 2019).



## 2.2 Preliminary Analysis of Wind Speed Data

Next we check if wind speed data are suitable for the reconstruction method just described. According to the rhs of equation (12) the estimation of the double conditioned pdfs $p(\Delta u_i|\Delta u_{i+1}, u^*)$ is necessary. However, to reduce computational costs or, respectively, the number of data points, it would be much more convenient to use the single conditioned pdfs $p(\Delta u_i|\Delta u_{i+1})$ by excluding the condition on the wind speed $u^*$ to be predicted (Nawroth et al., 2010; Peinke et al., 2019). Thus the equality

$$p(\Delta u_i|\Delta u_{i+1}; u^*) = p(\Delta u_i|\Delta u_{i+1}) \tag{13}$$

must hold. As it can be seen from fig. (1), eq. (13) holds for $u^* \approx 0$, but shows a significant shift for $u^* \approx 2.5$, therefrom we reason that the equality in equation (13) cannot generally be assumed, so we have to stick to the double conditioned pdfs $p(\Delta u_i|\Delta u_{i+1}, u^*)$. Similar results have been reported for idealised turbulence (Stresing and Peinke, 2010) and sea waves (Hadjihosseini et al., 2016). On derivation of our final formula for reconstruction of time series (cf. eq (12) an essential step was to premise the underlying scale process to be Markovian. Thus it has to been shown that for $\Delta\tau = \tau_{i+1} - \tau_i \geq \Delta\tau_{ME}$

$$p(\Delta u_i|\Delta u_{i+1}; ...; \Delta u_N; u^*) = p(\Delta u_i|\Delta u_{i+1}; u^*) \tag{14}$$

is a valid assumption. As the verification of this expression is not feasible in its generality, we limit ourself by reducing the number of dimension involved and just check the equality of

$$p(\Delta u_i|\Delta u_{i+1}; \Delta u_{i+2}; u^*) = p(\Delta u_i|\Delta u_{i+1}; u^*). \tag{15}$$

To check this we utilize Chapman-Kolmogorov equation (CKE) (Friedrich et al., 2011): The conditional pdf $p(\Delta u_i|\Delta u_{i+2}; u^*)$ is estimated directly from observational data and afterwards compared with the conditional pdf $\widetilde{p}(\Delta u_i|\Delta u_{i+2}; u^*)$ obtained numerically by use of the CKE

$$\tilde{p}(\Delta u_i|\Delta u_{i+2}; u^*) = \\ \int p(\Delta u_i|\Delta u_{i+1}; u^*)\, p(\Delta u_{i+1}|\Delta u_{i+2}; u^*), \tag{16}$$

whereas the two conditional pdfs within the integral on the rhs are as well directly estimated from data. Figure 2 shows that equation (16) only holds for $\Delta\tau \geq \Delta\tau_{ME}$. We find Markov-Einstein length $\Delta\tau_{ME} \leq 0.1\ s$, which we are going to use henceforth.

## 2.3 Parametrization of the Fokker-Planck-Equation

As it was mentioned in sec. (2.1) the FPE may be used to generate solutions for the needed conditional pdfs in eq. (1). Aiming for this, one needs a parametrization of the FPE reflecting the scale process of the real world wind speed data. No general, physical formulation for a FPE, describing the scale process of wind speeds is known, thus we use the possibility to estimate a parametrization directly from the given data (cf. eq. (9) and (N Reinke; Peinke et al., 2019)).

This way we get estimations of the drift and diffusion functions $D^{(1)}(\Delta u, \tau_i, u^*)$, $D^{(2)}(\Delta u, \tau_i, u^*)$ along the scale $\tau_i$ and for

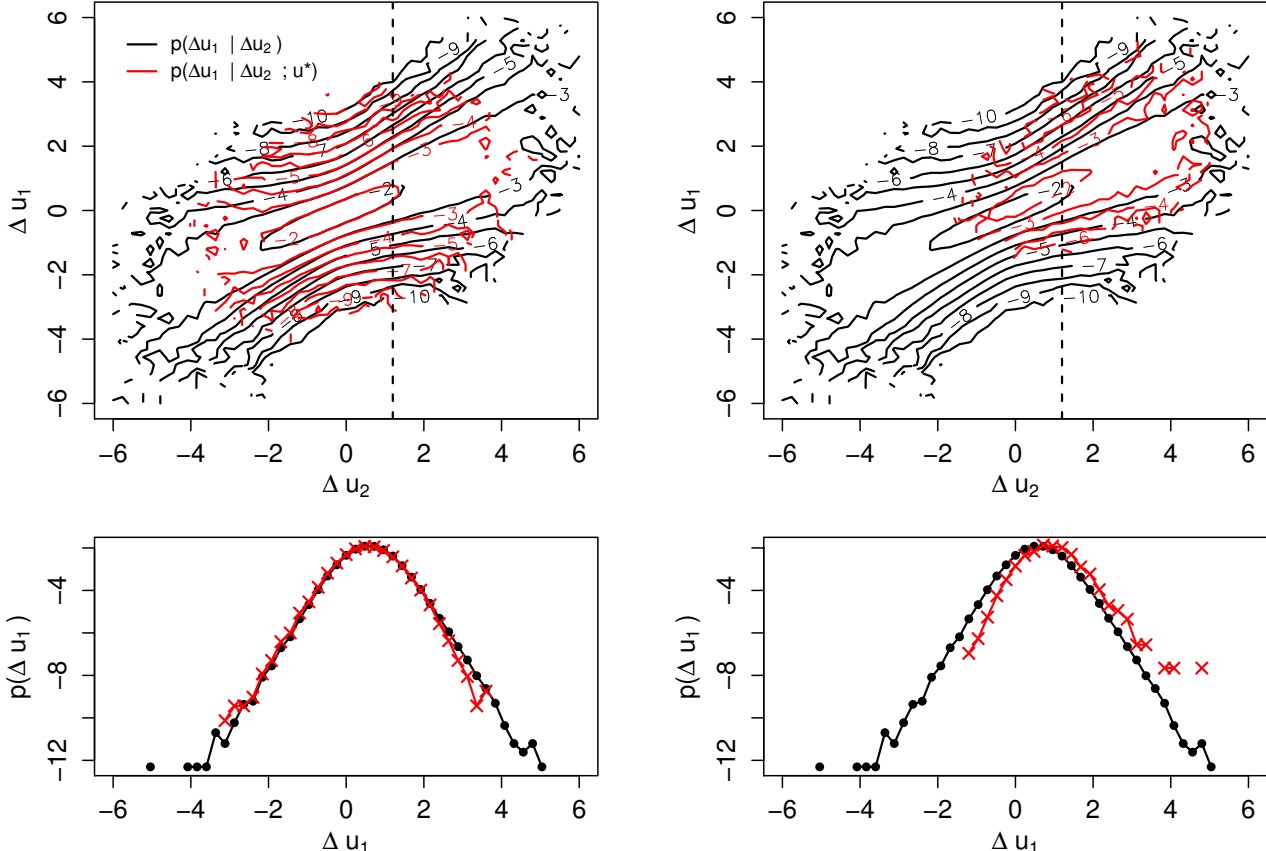

**Figure 1.** Comparison of single conditioned pdfs $p(\Delta u_1 | \Delta u_2)$ (black) and double conditioned pdfs $p(\Delta u_1 | \Delta u_2; u^*)$ (red) for $u^* \approx 0$ (left), $u^* \approx 2.5$ (right), $\tau_1 = 1s$ and $\tau_2 = 2\tau_1$. Dashed lines indicate cuts through the conditional pdfs at $\Delta u_2 \approx 1.2$. The levels of the contour plots are given in natural logarithmic scale.

every wind speed value $u^*$. From these estimations one then usually finds parametrization of the FPE by fitting appropriate polynomial surfaces to the estimated functions, which then may be used to obtain numerical solutions of the FPE.

155     To match the functional shape of the estimated $D^{(1)}(\Delta u, \tau_i, u^*)$ and $D^{(2)}(\Delta u, \tau_i, u^*)$ (see fig. (3)) we require the polynomials to be of order 3 and 2 respectively. We find a significant shift $\gamma_{D^{(1)}, D^{(2)}}(\tau_i, u^*)$ depending on $u^*$ and $\tau_i$ for both drift and diffusion along all scales $\tau_i$ which has to be taken into account for a parametrization suitable to the given data.



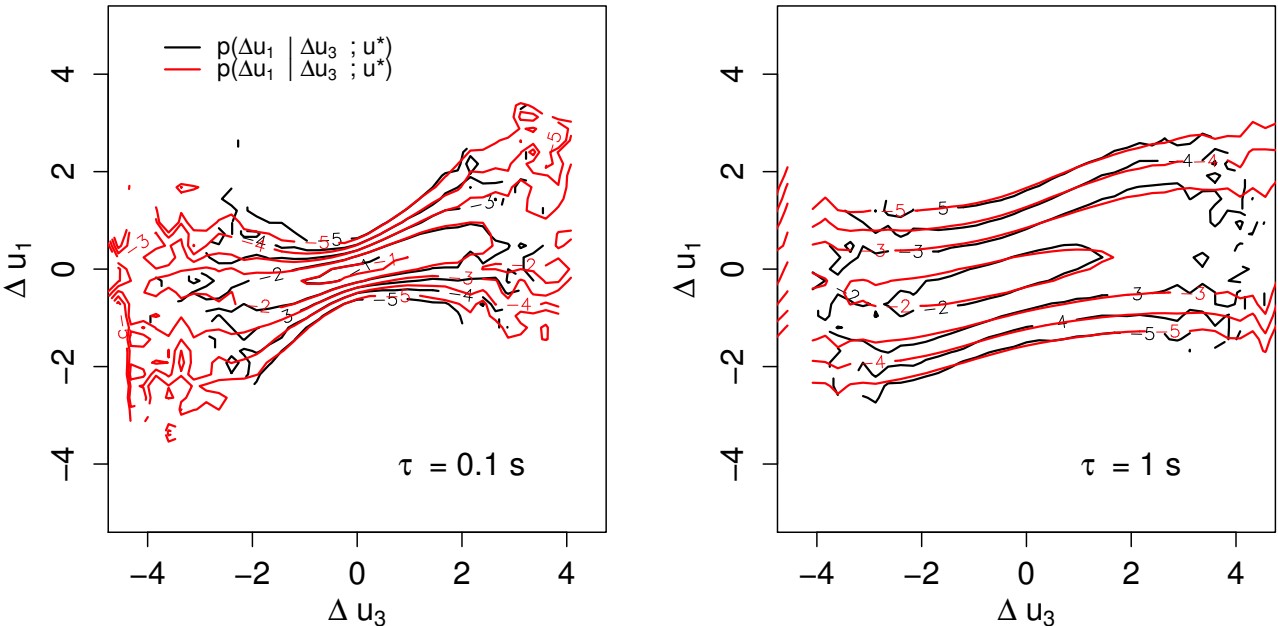

**Figure 2.** Comparison of the double conditioned pdfs $p(\Delta u_1|\Delta u_3; u^*)$ (estimated from data) and $\widetilde{p}(\Delta u_1|\Delta u_3; u^*)$ (obtained from CKE) for $\tau_1 = 0.1\ s$ (left) or $1\ s$ (right) and respectively $\tau_2 = 2\tau_1$, $\tau_3 = 3\tau_1$, $\Delta\tau = \tau_2 - \tau_1 = \tau_3 - \tau_2$ and $u^* = 0$.

We set for $D^{(1)}(\Delta u, \tau_i, u^*)$:

$$D^{(1)}(\Delta u, \tau_i, u^*) =$$
$$d_{10}(\tau_i, u^*) + d_{11}(\tau_i, u^*)\left[u - \gamma_{D^{(1)}}(\tau_i, u^*)\right] + \tag{17}$$
$$d_{13}(\tau_i, u^*)\left[u - \gamma_{D^{(1)}}(\tau_i, u^*)\right]^3$$

and for $D^{(2)}(\Delta u, \tau_i, u^*)$:

$$D^{(2)}(\Delta u, \tau_i, u^*) =$$
$$d_{20}(\tau_i, u^*) + d_{21}(\tau_i, u^*)\left[u - \gamma_{D^{(2)}}(\tau_i, u^*)\right] + \tag{18}$$
$$d_{22}(\tau_i, u^*)\left[u - \gamma_{D^{(2)}}(\tau_i, u^*)\right]^2.$$

Similar findings were made for other system (Hadjihosseini et al., 2016) and (Stresing and Peinke, 2010), where the dependence on $u^*$ was limited only to the drift function $D^{(1)}(\Delta u, \tau_i, u^*)$. For wind speed data the contribution of $u^*$ is clearly more complex.

Next, we check if the parametrization of the FPE, given by the eq. (17) and (18) performs well in describing the underlying scale process, before one uses it for the reconstruction scheme presented in sec. 2.1. This can be easily be done by comparing conditional pdfs estimated directly from the data and the ones obtained from numerical solutions of the FPE. The latter

160

165





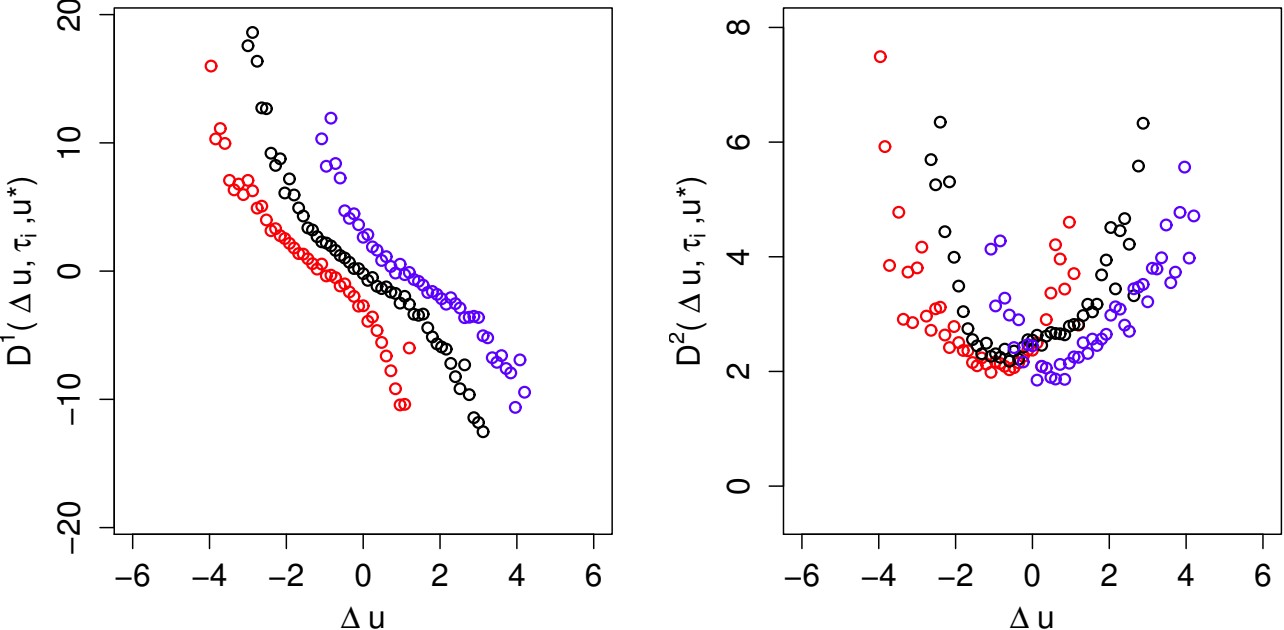

**Figure 3.** Exemplary estimations of the drift and diffusion functions $D^{(1)}(\Delta u, \tau_i, u^*)$, $D^{(2)}(\Delta u, \tau_i, u^*)$ for $\tau = 65\ s$ and $u^* \approx -1.58$ (red), $\approx 0$ (black) and $\approx 1.58$ (blue).

one is not carried out by common finite-difference schemes, but by an iterative approach (for first works see (Renner et al., 2001),(Waechter et al., 2003)). For an (very) small step size in scale $\Delta\tau$ the functions $D^{(1)}(\Delta u, \tau_i, u^*)$, $D^{(2)}(\Delta u, \tau_i, u^*)$ can assumed to be constant in $\tau$, leading to an exact solution (cf. (Risken, 1996)) for the conditional pdf

$$
\begin{aligned}
p(\Delta u_j, \tau_i - \Delta\tau | \Delta u_i, \tau_i, u^*) &= \frac{1}{2\sqrt{\pi D^{(2)}(\tau_i, u^*)\Delta\tau}} \cdot \\
&exp\left[ -\frac{\left(\Delta u_j - \Delta u_i - D^{(1)}(\tau_i, u^*)\Delta\tau\right)^2}{4D^{(2)}(\tau_i, u^*)\Delta\tau} \right],
\end{aligned}
\tag{19}
$$

describing the transition between wind speed increments of a larger scale $\tau_i$ to a smaller scale $\tau_i - \Delta\tau$. By iteratively combining this so called *short time propagator* (STP) with the CKE (see eq. 16) one is able to obtain conditional pdfs $p(\Delta u_j, \tau_j | \Delta u_i, \tau_i, u^*)$ for arbitrary large scale differences $\tau_i - \tau_j \gg \Delta\tau$. In theory this procedure would lead to exact solutions of the FPE, but since one is limited to finite step sizes $\Delta\tau$, this methods of course only provides numerical approximations of the true conditional pdfs.

Comparing conditional pdfs estimated directly from the data and from the numerical solution we see (fig. 4) that our proposed estimation in terms of $D^{(1)}(\Delta u, \tau_i, u^*)$ and $D^{(2)}(\Delta u, \tau_i, u^*)$ is well suited to describe the underlying scale process. Thus we use this parametrization for the reconstruction scheme.





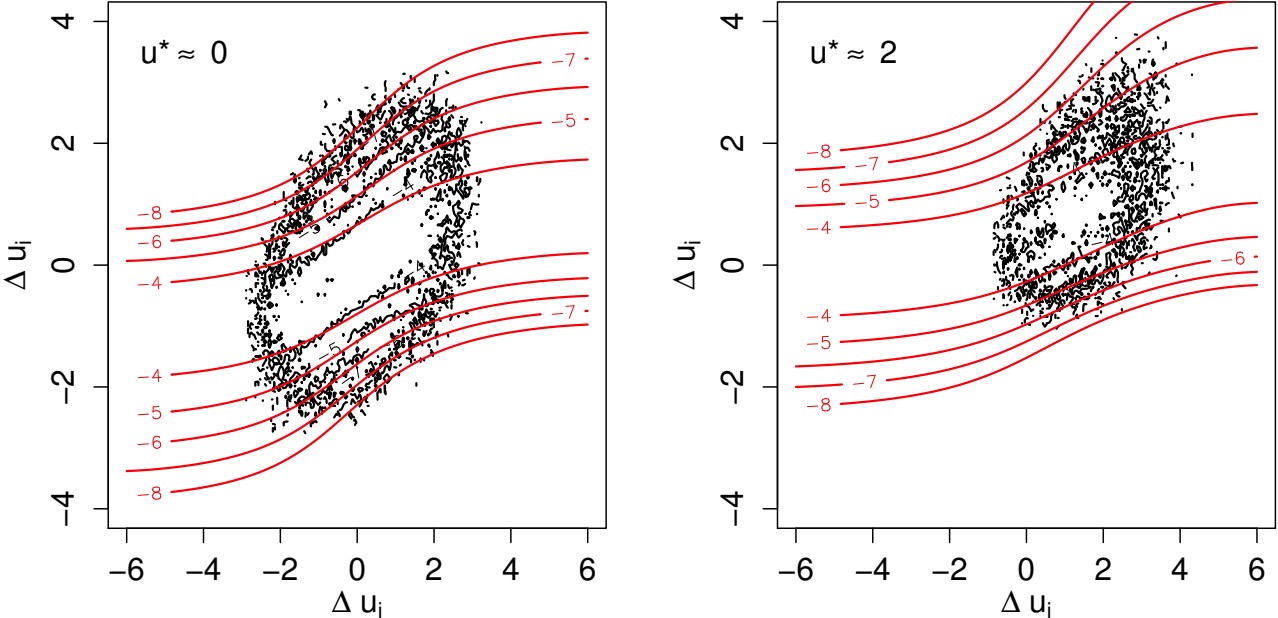

**Figure 4.** Isoline plots of a conditional pdf $p(\Delta u_i, \tau_i | \Delta u_j, \tau_j; u^*)$ estimated from the data (black) and from numerical solution of the FPE (red) for $\tau_i = 1.6\ s$, $\tau_j = 3.2\ s$ and for $u^* \approx 0$ and $u^* \approx 2$

## 2.4 Results of the Multiscale Reconstruction

Alternatively to the presented approach to obtain the conditional pdfs from numerical solutions of the FPE, it is of course possible and much less cumbersome to estimate them directly from observational data. (Note the use of the FPE is less noisy and extends to large values as seen in fig. 4.) In this section we will present the results for the multipoint reconstruction achieved for $u^*$, yielding from both approaches.

To start the reconstruction scheme, we provided a short piece of the original time series of $N$ wind speeds to the algorithm to compute the pdf $p(u^* | u_1, ..., u_N)$, from which the first point of our simulation can be drawn. The reconstruction scheme is then shifted by one time step $\tau$ and applied again. By iteratively applying our method a new artificial time series of arbitrary length can be generated. After $N$ iterations of the reconstruction scheme no data from measurements are required anymore to generate new values of $u^*$. From visual comparison in fig. 5 one finds a realistic looking simulated time series of $u^*$, retaining the characteristics of dynamics on the smaller scales as well as the ones of the larger scales. To confirm this visual impression in a quantitative way, we compare the increment pdfs $p(\Delta u_i, \tau_i)$ obtained from the reconstructed and the measured time series. The synthetic time series were generated by using both, the conditional pdfs directly estimated from data and from numerical solutions of the PFE. As shown in fig. 6 the increment pdfs yielding from both stochastic simulations nicely coincide with the pdfs from observations. This attests that the presented reconstruction scheme is able to capture the complex dynamics of

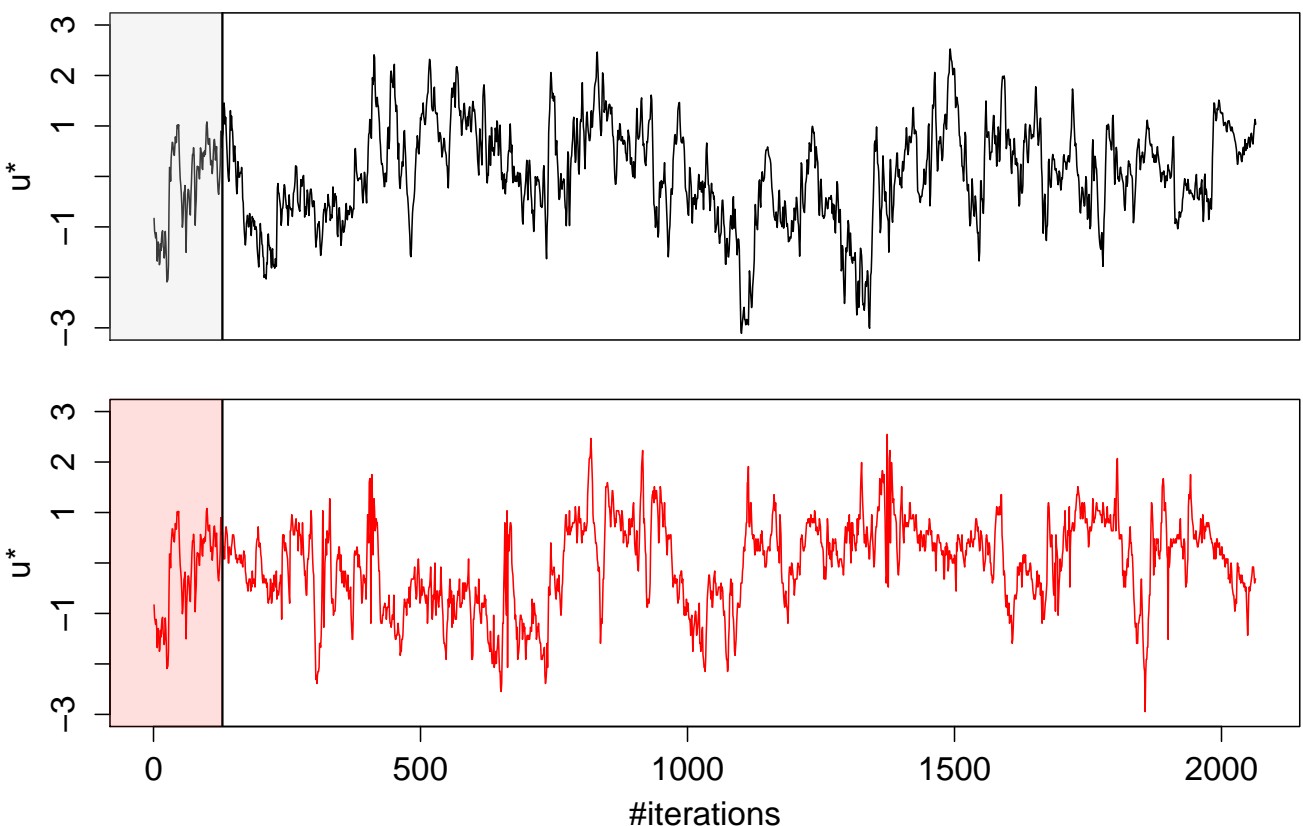

**Figure 5.** Comparison of original wind speed time series $u^*$ (black, upper picture) with the reconstructed one using eq. 13 (red, lower picture). The vertical line marks the transition from the $N = 128$ on a logarithmic scale points provided as starting value for the reconstruction scheme to the simulated wind speeds.

wind speeds, characterized by a gradual shift of increment pdfs of a Gaussian-like shape (larger scales) to increment pdfs of heavy-tailed shape (smaller scales).

A striking difference between the increment pdfs from the stochastic simulation can be noted: Whereas the tails of the original pdfs are systematically underestimated by the reconstruction using the directly estimated pdfs, the reconstruction from the numerically obtained conditional pdfs is able to keep track of the tails of the original pdf. This stems from the fact, mentioned above, that by estimating a pdf from observational data one in general underestimates the outer tails, since there are only few measured points available. Considering the estimation of conditional pdfs the estimation error of course worsens, which is even more severe when additional conditioning is applied, like in our case in terms of the additional condition on $u^*$.

The tails of conditional pdfs $p(\Delta u_i, \tau_i | \Delta u_j, \tau_j; u^*)$ computed from the family of FPEs can be extrapolated to areas where no measurement points are available. This effect is a direct consequence of the CKE (16), as the tails are the product of quite well





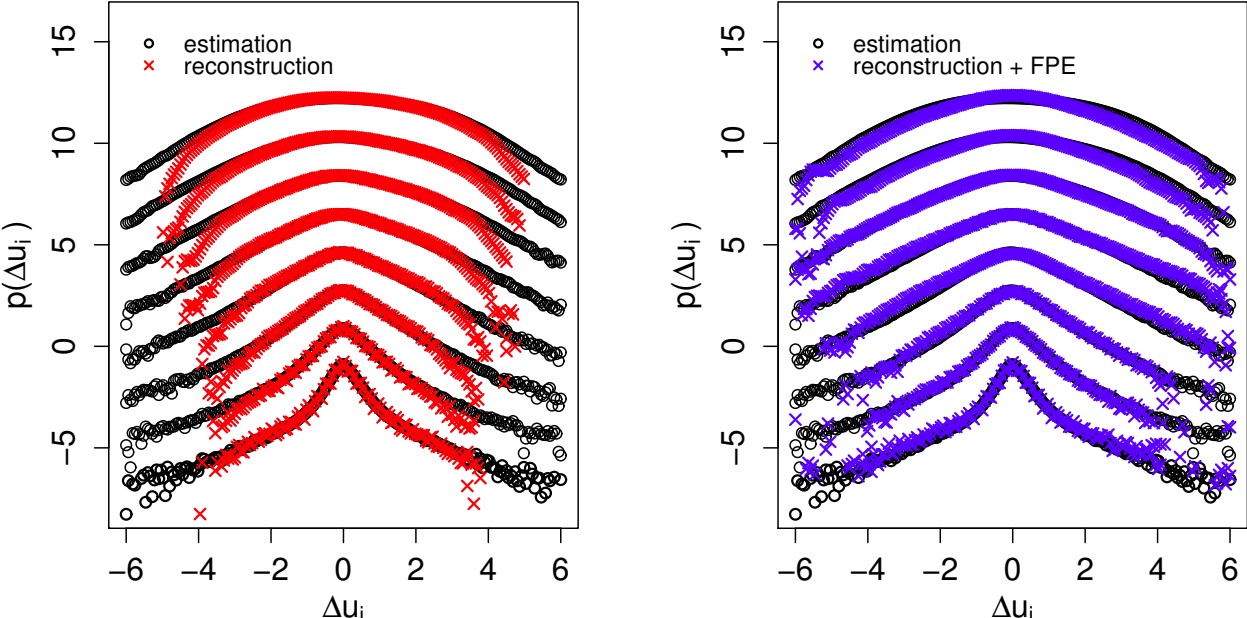

**Figure 6.** Comparison of the marginal increment pdfs computed from the empirical data (black), the simulated data using the directly estimated pdfs (red) and the simulated data using the conditional pdfs obtained from numerical solution of the FPE (blue). The scales range from $\Delta\tau_{ME}$ to $N \cdot \tau_{ME}$, they are explicitly: $2^i \cdot \Delta\tau_{ME}$ with $(i = 0, 1, ..., 7)$ and $\Delta\tau_{ME} = 0.1\ s$. For better visualization the pdfs were shifted along the vertical axis.

estimated less probable but not rare events.

Certainly this approach indirectly suffers from the limited number of observations as well, as the estimation of the function $D^{(1)}(u, r, u^*)$ and $D^{(2)}(u, r, u^*)$ is based on observational data, too (cf. eq. 9). Furthermore the parametrization of these functions is always only a approximation of the real drift and diffusion functions, introducing deviations from the real world system.

But we see from fig. 6) that the majority of increments are well grasped, only the occurrence of a few rare events are under predicted. A more detailed investigation of the pdfs shows the pdfs obtained by the FPE deviates from the original shape for the largest scale $\tau = N \cdot \Delta\tau_{ME}$, but is performing better in the outer wings of the pdf.

From fig. 7 a better understanding of the reconstruction method can be gained: The pdf used for the simulation $p(u^*|u_1, ..., u_N)$ is not stationary, even though it is computed from completely stationary conditional pdfs (cf. 13) and changes sensitively with respect to the $N$ past values. While the snapshot pdf (shown as black circles) at the time marked by the black vertical line has a rather clear shape, it undergoes a change, becoming broader (red line and red circles). Due to the spreading of the wings of the pdf, values of $u^*$ of larger magnitude become more likely to be drawn. This may lead to a distinct shift of the wind speed values to $u^* \leq 0$ as seen in this example. After this transition the broadness decreases (blue line and circles) again and a tendency to

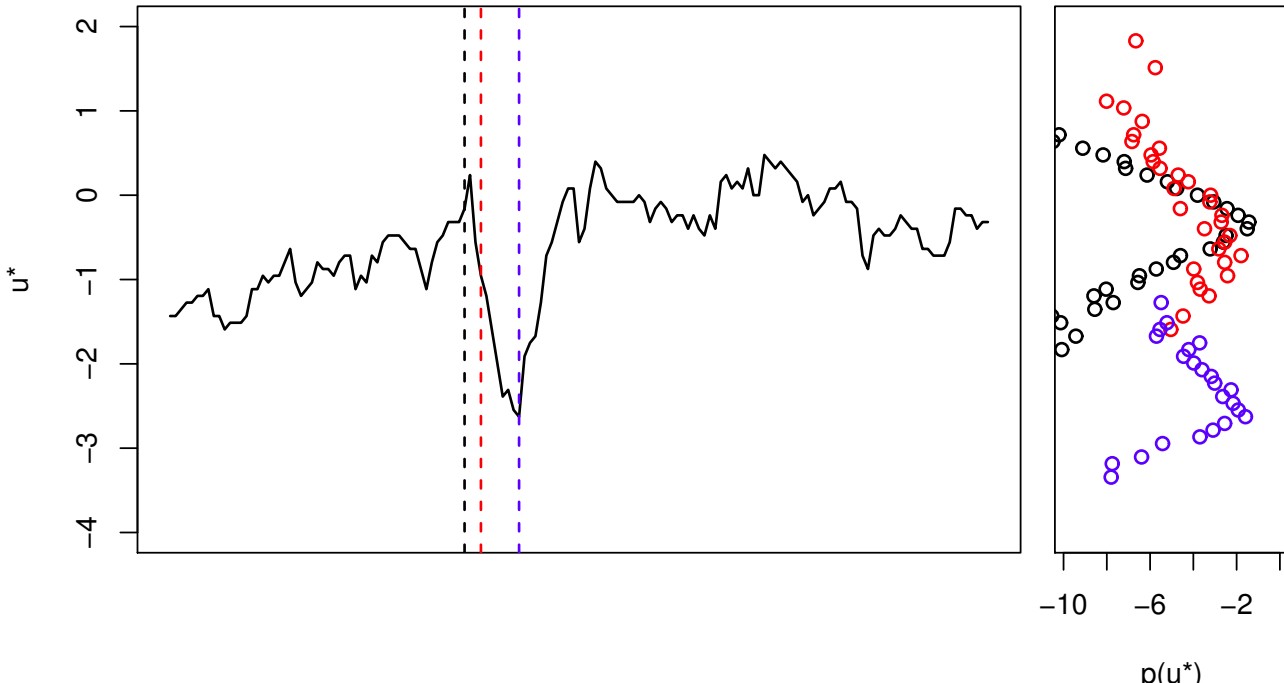

**Figure 7.** Evoluion of $p(u^*|u_1,...,u_N)$ during reconstruction. Horizontal lines indicate snapshots of the pdf used for drawing the next sample of $u^*$. The colors of the horizontal lines respectively correspond to the snapshot pdfs.

relax back to $u^* = 0$ can be seen. Furthermore the increased broadness of the red pdf (red line) can be seen as an early warning signal that the wind speed is prone to fluctuate in a stronger way.

## 3    Extension to non-stationary wind speeds

In the preceeding part for the reconstruction sheme block-wise normalized wind speeds with a window length of $1\ min$ were used. These blocks were defined by common mean wind speed $\overline{U}$ and standard deviation $\sigma_U$. For the normalised wind speed $u$

we showed how to generate new time series see Fig. 8.

There are different methods to generate more general non-stationary wind data. Knowing the slow variation of $\overline{U}(t)$ and $\sigma_U(t)$ the drift and diffusion coefficients $D^{(i)}$ are taken as slowly changing function of $D^{(i)}(\Delta u, \tau, u^*, \overline{U}, sigma_U)$. If due to the normalisation of $U$ to $u$ the coefficients $D^{(i)}$ are in a good approximation independent on $\overline{U}$ and $\sigma_U$, the slow variation of the real wind conditions can be incorporated over the backtransformation of the newly estimated values $U^* = (\sigma_U \cdot u^*) + \overline{U}$. The





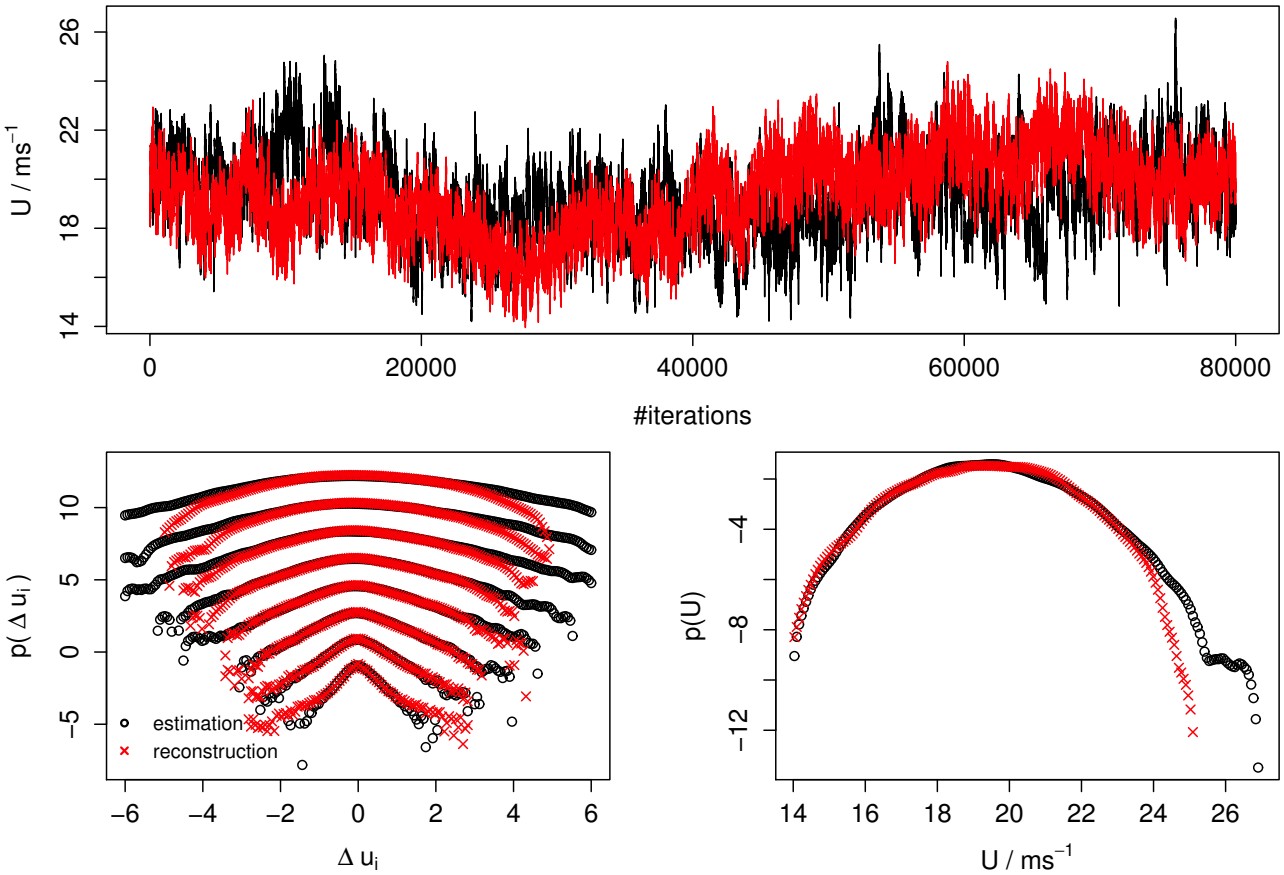

**Figure 8.** Upper panel: comparison between measured non-stationary speeds $U$ and reconstructed non-stationary wind speeds. Lower panel: increment (left) and one point (right) pdfs of original and reconstructed $U$.

slow dynamics of $\overline{U}(t)$ and $\sigma_U(t)$ may be given from measured data, meteorological simulations or other modelling.

A third possibility is a self-adaptive procedure which we show here. Instead of using given values of $\overline{U}(t)$ and $\sigma_U(t)$, the intrinsic fluctuation of these quantities are used: Given an initial pair $(\overline{U}(t), \sigma_U(t))$ estimated over $1\ min$ from measured data, a time series of non-stationary wind-speeds $U^*(t)$ is obtained by applying the above mentioned backtransformation to the

generated wind speeds $u^*$ from our algorithm. For the upcoming simulation window of $1\ min$ length a new pair $(\overline{U}(t'), \sigma_U(t'))$ is estimated from the just generated block of wind speed data $U^*(t)$. This procedure is carried out until a time series of non-stationary wind speeds of desired length is obtained. In fig. 8 such a time series is shown together with the statistical analysis of the increment pdfs and and the marginal pdf of the non-stationary wind speed $U$. We observe a fairly nice match between the empirically and reconstructed pdfs. At this point we would like to emphasize that we do not aim to create copies of historical

wind speeds, but to be able to generate stochastic equivalent time series.



## 4    Conclusions

We presented a stochastic approach based on multipoint statistics to generate surrogate short time wind speed fluctuations with stochastic processes. Note these stochastic processes can be estimated self-consistently from given data. By using the normalized wind speeds $u^*$ and wind speed increments $\Delta u(\tau_i)$, $\Delta u(\tau_j)$ from two separate scales $\tau_i$ and $\tau_j$ a three-point closure to

the complex systems of wind speeds was achieved.

It was shown that our method works well in describing the dynamics of block-wise normalized wind speeds $u^*$ along scales $\tau_i$ in terms of a stochastic scales process, governed by a family of Fokker-Planck equations. This separation of the fluctuations from the mean values is similar to *RANS*-approach widely used in fluid dynamics (Frisch and Kolmogorov, 1995), with the great difference that we have a description of the underlying stochastic process of the fluctuation and 'only' lack the mean

values. With the modified reconstruction (cf. sec. 3) we are able to generate mean values on basis of past values of the reconstructed time series, yielding realistic non-stationary wind speed time series $U^*$. As the typical response times of wind turbines and their control systems have duration of seconds to minutes, our reconstructed wind data are suitable for investigation of many dynamical effects of the wind energy conversion process. Note for these times the wind energy system may be driven in non-stationary response dynamics. Thus for these time scales it is important to have to maximal information on all statistical

details of the driving wind source. The stochastic model presented here is based on multipoint statistics and thus captures small scale intermittency effects, extreme events as well as clustering of fluctuations, up to not addressed in wind energy research.

For time scales larger than the response times of wind turbines, the turbines operate with fully adapted control systems in a stationary state. To estimate effects, like e.g. loads, of such stationary states the temporal order of the states becomes unimportant. It is sufficient to know how often which wind situation emerges. Thus the knowledge of the valid Weibull distribution

$p(\overline{U})$ should be sufficient. Note, our result here indicates that it would be better to extend the Weibull distribution to the joint probability $p(\overline{U}, \sigma_U)$.

Finally we emphasise that the presented stochastic multipoint approach to small scale wind speed fluctuations should encompass automatically extreme short term wind fluctuation, commonly added to investigation in term of standard one or multi-year gusts. This methods can be applied easily to other wind quantities like the temporal behaviour of shears, or wind veers, even-

tually combined in higher dimensional stochastic processes (SIEFERT and PEINKE, 2006). The results reported in (Ali et al., 2019) show that such a stoachastic modelling can also be used for wake flows.

**acknowledgement:** We acknowledge the funding of this project by the VolkswagenStiftung grant and the support from the Ministry for Science and Culture of the German Federal State of Lower Saxony (Grant No. ZN3045, nieders. Vorab to M. W.). Furthermore we acknowledge helpful discussions with Andeé Fuchs and Hauke Hähne.



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
