# Peer review of "Multipoint Reconstruction of Wind Speeds"

_Wind Energy Science, 2020_

## Referee Comment (RC1) · Anonymous Referee #1 · 6 Mar 2020

In their manuscript 'Multipoint Reconstruction of Wind Speeds', the authors develop a multi-point stochastic algorithm that is capable to generate synthetic wind speed time series. This is similarly done via the use of field data and the support of a Fokker-Planck framework alike. The method presented is sound, and applications are wide-ranging in the design and analysis of wind farms and fields. As an extension, the authors also deal with the non-stationarity that is typical for real-world wind speed fluctuations. Overall the paper is a welcome addition to the literature and shows promising results. However, its technical presentation doesn't give the promising results enough credit and is dragging the quality of the manuscript down. I would not suggest publication in its current state, but with a bit of additional effort, this should be redeemable. Concerning this, I do have a series of comments on the manuscript that should be ad-

dressed. Once they have been properly answered, I would suggest the manuscript for publication.

General Comments:

Overall language and grammar: The comma placement, hyphenation, and grammar need improvements. The authors should give the text another round of fixes to ensure the proper use of language.

Section 1 through 2.1: In the introduction and methods section, the authors build upon the senior's authors work in Nawroth et al (2006) for turbulent flows. The authors do give the reference, but it should be made clearer which parts are new and which parts are a simple reproduction.

Font sizes and font styles are inconsistent between the figures and the text. This should be fixed.

Specific Comments: Page 1, line 2: The authors introduce the abbreviation 'cpdfs', but inconsistently use 'conditional pdf' (e.g. Page 3, line 86) throughout the paper. This should be consistent.

Page 2, line 28: Define what 'short time scales' means and how it relates to intermittency.

Page 2, line 35: How is the complexity of the wind energy conversion process connected to the desire of finding 3D velocity fields?

Page 3, line 60: '1 min' should read '1 minute long'.

Page 3, line 60: Even though $U(t)$ is an obvious reference to a velocity, it should be defined in the text (especially as to see whether it is the mean of the full 3D velocity vector or of one component).

Page 3, line 60ff: How much data was used?

Page 3, line 64f: It should be spelled out what these abbreviations are meant to abbreviate.

Page 3, line 65f: Is this normalization justified? There could potentially a strong coupling between the mean and the standard deviation that erroneously gets averaged by this procedure. The authors should provide a plot of standard deviation vs mean for their dataset to build confidence in this grouping of blocks.

Page 3, line 68: Explain how the wind speeds emerge from a turbulent cascade.

Page 3, line 78: Taylors frozen flow hypothesis only holds if the fluctuations are small compared to the mean flow. Is this the case here? Provide numbers.

Page 3, line 86: The abbreviation 'lhs' (and 'rhs') should be defined, even if it is commonly used.

Page 4: The derivation appears unwieldy (and is a reproduction of previous work) and might be better suited for supplemental materials.

Page 4, line 102: The N. Reinke reference is missing a year.

Page 5, line 105: The citation style for Risken 1996 is different from other references.

Page 5, line 109: The argument on the minus sign seems handwaving. Even though it is commonly repeated, a more rigorous, mathematical explanation would be desirable.

Page 5, 110: Provide a reference for the Pawula theorem.

Page 5, line 111: Provide a plot in the manuscript to show that this approximation is valid for the data at hand.

Page 5, line 118: I disagree with the statement that 'it can be easily seen'. Provide more details.

Page 6, line 139: The authors only check the validity of equation 15 for a single choice of the difference time tau = 1s. Given the potential of long-range correlations or eddies

in the flow, this expression should be checked for several choices of tau.

Page 7, figure 1: The isoline heights are hard to read.

Page 7, line 156: Introduce abbreviations.

Page 8, line 160: What is $d_{10}$, etc?

Page 9, figure 3: On Page 7, line 156 the authors write that the drift and diffusion terms should be polynomials of order 2 and 3. Provide fits in figure 3 to illustrate this.

Page 9, equation 19: Do not put 'exp' in italics.

Page 10, figure 4: The black-on-black isoline notations are virtually impossible to read.

Page 10, figure 4: The math in the right panel does not look convincing. Can the authors comment on this?

Page 13, line 227: Typo: replace sigma with \sigma.

Page 13, line 227: Why can the coefficients D be considered slowly changing functions? Figure 7 seems to show the contrary.

Page 14, figure 8: Consider splitting the top panel into two. Through the overlapping curves, a lot of detailed information is getting lost.

Page 15, line 248: RANS should be spelled out.

Page 15, line 249: 'Great' is an odd choice of word.

Page 15, line 255: Consider rephrasing the sentence. Not all multipoint-based models automatically capture small-scale intermittency.

Page 15, line 265: Do not use capitals in the reference.

Page 15, line 267: Capital 'A' in acknowledgments.

Page 15, line 269: Is the first name of the person Andeé?

Page 16 References: A lot of the references are inconsistent when it comes to providing doi, placement of first name letters and abbreviation dots.

---

## Referee Comment (RC2) · Anonymous Referee #2 · 17 Mar 2020

This paper presents a methodology for generating wind speed fluctuations. The methodology is well described, and the paper is overall well written. However, this reviewer has a list of specific comments, including some major comments, that should be addressed before this manuscript can be accepted for publishing.

Specific comments:

1. L13. I would argue that hydro is more represented in decarbonized energy sources than wind and solar; at least in some parts of the world. It should be included in this list.
2. L15. Why wind(solar) is capital in the manuscript?
3. L25. The citation style is incorrect. Please revise accordingly.
4. L28. Should be "…known to be…"
5. L33. There should be a space between 10 and min. Please apply the same correction everywhere else (number and unit separation). Also, the citation style is incorrect. Please revise this issue everywhere in the manuscript.
6. The exact definition of intermittency (for the context of this study) should be provided in the Introduction. The authors talk a lot about intermittency, but the exact definition is not provided.
7. L46. Remove one "and" at the end of this line.
8. L60. It should be specified that t is the time.
9. L61–62. Please revise the sentence for proper English.
10. L60 and L65. Please clarify the difference between u(t) and U(t).
11. The abbreviation pdf is sometimes italicized and sometimes not. Please be consistent.
12. L110. There should be a comma after the Pawula theorem. Also, please provide a reference for this claim on L110 and L111.
13. L155–156. Why the order of the polynomial of 3 and 2. Is this the lowest polynomial order that properly fits the data?
14. Figure 2. The two labels in the legend are identical, but the different notation is used in the figure caption. Please correct this before this figure can be reviewed properly.
15. L167. Please correct the English.
16. All figures. Please add (a), (b), (c), etc. labels for subplots.
17. L179. I belie that "an" should be "a".
18. Equation 19. The function exp should not be italicized. The same holds for any other function in the manuscript.
19. L223. The word min should not be italicized.
20. MAJOR COMMENT: L226. In non-stationary wind speed records, the fluctuations are dependent on wind speed. Reading this section (and this particular line), this reviewer concludes that the presented methodology does not account for this relationship. For instance, in the case of non-stationary thunderstorm winds, Chen and Letchford (2004) (doi: 10.1016/j.engstruct.2003.12.009) modulated the fluctuations based on the moving-mean wind speed. A similar approach was used by Chay et al. (2004) (doi: 10.1016/j.engstruct.2005.07.007). This has been shown on the example of full-scale data of thunderstorm winds in Burlando et al. (2017) (doi: 10.1175/MWR-D-17-0018.1) and Zhang et al. (2018) (doi: 10.1016/j.probengmech.2017.06.003). Notice that in these papers the moving-mean turbulence intensity in the transient (thunderstorm) wind record is not changing in time. This confirms that

the fluctuations increase as the mean wind speed increases. Please clarify this issue because it is particularly important for transient wind speed records.

21. This change (previous comment) would perhaps correct for the discrepancies between the measurements and the reconstruction in Figure 8 (pdfs).

22. MAJOR COMMENT: Related to my previous comment, non-stationary velocity records are often non-Gaussian too. Can you please clarify how is this accounted for in your methodology?

23. MAJOR COMMENT: The purpose of this methodology is to generate fluctuating wind records. This topic addressed in the seminal paper by Shinozuka (1972) (doi: https://doi.org/10.1016/0045-7949(72)90043-0). Without going into mathematical rigor in this review, the basis of his method is to generate random numbers (through Monte Carlo) that follow the prescribed power spectral density of wind fluctuation (e.g., Kamal spectra, Davenport spectra, von Karman spectra, Mann spectra, etc.). This method is later implemented in some of the studies provided in my comment 20 and references therein. So, my question is how the method proposed in your study extends beyond this well-established methodology of generating wind fluctuations? What are the benefits of using the presented method in your study?

24. MAJOR COMMENT: Can the authors plot spectra of the two velocity time series in Figure 5? Please also include the reference $-5/3$ slope for benchmarking.

25. MAJOR COMMENT: Going back to L36 in the Introduction. The authors correctly talk about the spatial dependency of fluctuations and coherence. How is the current model generating fluctuations in space? The presented results are for a point measurement, but the implementation for wind energy (i.e., wind turbine) analysis requires the spatially dependent profile. How a coherence function can be implemented in the method?

26. L238. The phrase "a fairly nice match" is not scientific. Please be specific.

27. How computationally efficient is your method? How much computational time is required to generate a fluctuation time series of different lengths? Can you please comment on this?

28. L252–L253. Not necessarily until the method accounts for the spatially coherent fluctuations.

29. References. Some citations include article titles while the others do not. In addition, some journal names are abbreviated whereas the others are not. Please be consistent.

30. Title: what exactly the authors mean by "multipoint?" This reviewer assumes this word signifies the time dependency of the methodology. If yes, isn't this redundant because fluctuations have to be time dependent?

---

## Author Comment (AC1) · 2 Jun 2020

**Reply to anonymous Referee No.1: Multipoint Reconstruction of Wind Speeds**

Christian Behnken, Matthias Wächter, and Joachim Peinke

Institute of Physics/ForWind, University of Oldenburg.

**Correspondence:** Peinke (peinke@uni-oldenburg.de)

*Copyright statement.* TEXT

**NOTE: All figure and equation numbers refer to the original submitted manuscript and may differ from the ones in the revised version.**

**Page 1, line 2: The authors introduce the abbreviation 'cpdfs', but inconsistently use 'conditional pdf' (e.g. Page 3, line 86) throughout the paper. This should be consistent.**

Changed in the revised manuscript.

**Page 2, line 28: Define what 'short time scales' means and how it relates to intermit- tency.**

We added following explanation to the revised manuscript:

"With short time scales we refer to time scales in the range of seconds to minutes. As it can be seen in (Boettcher et al., 2003). The effect of intermittency is most prominent at time scales $< 1$ s, but as the time scales increase, the pdfs broaden."

**Page 2, line 35: How is the complexity of the wind energy conversion process con- nected to the desire of finding 3D velocity fields?**

To avoid misunderstanding we reformulated our sentence in the manuscript:

"The missing of the basic understanding, the impracticability of handling such huge data sets as well as the complexity of the wind energy conversion process leads often to the demand of simplified models for wind speed."

in a new way:

"The impracticability of having all details of a turbulent wind field leads to the demand of the praxis to have access to simplified models for wind speed."

**Page 3, line 60: '1 min' should read '1 minute long'.**

Changed accordingly.

**Page 3, line 60: Even though U(t) is an obvious reference to a velocity, it should be defined in the text (especially as to see whether it is the mean of the full 3D velocity vector or of one component).**

We added a definition in the revised manuscript: "With $U(t)$ we refer to the resulting wind speed from the horizontal components."

**Page 3, line 60ff: How much data was used?**

We added the information in the manuscript: "The data were recorded at a sampling frequency of 1 Hz between calender week 1 to 10 in 2007 with an ultrasonic anemometer, mounted at 80 m height, resulting in approximately $6 \cdot 10^6$ samples."

**Page 3, line 64f: It should be spelled out what these abbreviations are meant to abbreviate.**

Changed accordingly.

**Page 3, line 65f: Is this normalization justified? There could potentially a strong coupling between the mean and the standard deviation that erroneously gets averaged by this procedure. The authors should provide a plot of standard deviation vs mean for their dataset to build confidence in this grouping of blocks.**

Indeed there is a significant coupling between mean wind speeds and fluctuations. Measuring the strength of the fluctuations by means of the standard derivation, we detect a quadratic scaling between both quantities fig. 1). However we aim at modelling the quasi-stationary wind speed fluctuations $u^*(t)$, which are obtained by a blockwise normalization (of 1 min length) with respect to the mean and standard derivation of wind speeds $U(t)$. This way we decouple the fluctuations from the magnitude of the mean flow. As stated later on, a rescaling of the fluctuations is achieved when we transform the modelled fluctions $u^*(t)$
back to real wind speeds $U^*$, by multiplying it with the standard derivation: $U^* = (\sigma_U \cdot u^*) + U$.

[Figure]

**Figure 1.** dependency of standard derivation $\sigma_U$ on mean flow $\overline{U}$

**Page 3, line 68: Explain how the wind speeds emerge from a turbulent cascade.**

It is well known that the turbulent behavior of wind speed below 10 min becomes similar to the ideal homogeneous isotropic turbulence involving a turbulent cascades from large to small scales. The common arguments are that below 10 min the wind turbulence becomes three dimensional whereas for larger time scales the turbulence is more like a two dimensional turbulence with a cascade to larger scales. This transition between two and three dimensional turbulences with different directions of the cascade, in which the energy is transported, supports the idea of the similarity between three dimensional homogeneous isotropic turbulence and wind turbulence on smaller scales. A statistical analysis of wind data supporting this results are given in (Morales et al., 2012).

**Page 3, line 78: Taylors frozen flow hypothesis only holds if the fluctuations are small compared to the mean flow. Is this the case here? Provide numbers.**

As one can see from the provided histogram of the fractions of the fluctuations $u^*$ and the mean flow $\overline{U}$ (cf. fig. (2)), in most of the samples the fluctions are at least one magnitude smaller than the mean flow. In accordance we could add to our paper the comment that we use Taylors frozen flow hypothesis as an approximation, which might be improved in future.
However our statement about Taylos frozen flow hypothesis was just supposed to be a general remark regarding the term 'multi-point' in the context of time series. As we are dealing with time series here, there is no check the validity Taylors hypothesis in our paper.

[Figure]

**Figure 2.** dependency of standard derivation $\sigma_U$ on mean flow $\overline{U}$

**Page 3, line 86: The abbreviation 'lhs' (and 'rhs') should be defined, even if it is commonly used.**

Changed accordingly.

**Page 4: The derivation appears unwieldy (and is a reproduction of previous work) and might be better suited for supplemental materials.**

We removed this part of the method section accordingly and shifted it to the appendix.

**Page 4, line 102: The N. Reinke reference is missing a year.**

Changed accordingly.

**Page 5, line 105: The citation style for Risken 1996 is different from other references.**

Changed accordingly.

**Page 5, line 109: The argument on the minus signseems handwaving. Even though it is commonly repeated, a more rigorous, mathematical explanation would be desirable.**

The minus (eq. (8)) is motivated by the fact the we consider the turbulent cascade going from large to small scales. With loss of generality a positive sign may be used (see (Peinke et al., 2019)).

**Page 5, 110: Provide a reference for the Pawula theorem.**

We added the reference in the revised manuscript (Risken 1996).

**Page 5, line 111: Provide a plot in the manuscript to show that this approximation is valid for the data at hand.**

We added a plot in the the revised manuscript: "As one can see in fig. (3), the fourth Kramery-Moyal coefficient is slightly larger than zero, but negligible compared to the magnitude of the diffusion function $D^{(2)}$"

[Figure]

**Figure 3.** Exemplary estimations of the second and fourth Kramers-Moyal coefficient $D^{(2)}$ and $D^{(4)}$ for $\tau = 65\ s$

**Page 5, line 118: I disagree with the statement that 'it can be easily seen'. Provide more details.**

We added are more detailed explanation in the revised manuscript.

**Page 6, line 139: The authors only check the validity of equation 15 for a single choice of the difference time tau = 1s. Given the potential of long-range correlations or eddies in the flow, this expression should be checked for several choices of tau.**

The validity of equation 15 has been checked for a time scale separation $\Delta\tau_{EM} = 0.1\ s$, being the Einstein-Markov length for this data set, and as well for $\Delta\tau > 0.1\ s$, namely $\Delta\tau = 10 \cdot \Delta\tau_{EM} = 1.0\ s$ (see fig. 2) shown within our paper. As described in the beginning of the method section, the wind speeds $U(t)$ are transformed to quasi-stationary wind speed fluctuations $u(t)$, thus long-range correlations (longer that 1 min) will be eliminated by that procudure.

**Page 7, figure 1: The isoline heights are hard to read.**

Changed accordingly.

**Page 7, line 156: Introduce abbreviations.**

Added accordingly.

**Page 8, line 160: What is $d_{10}$, etc?**

The polynomial coefficients $d_{ij}(\tau_i, u^*)$ are themselves higher-order polynomial obtained from fitting the the third and second order polynomial coeffients over all considered $\tau_i$ and $u^*$. We explicitly obtained following polynomials (and added them to the appendix of the revised manuscript):

$$d_{10} = c_{0,d_{10}} \cdot \tau_i + c_{1,d_{10}} \cdot u^* \cdot \tau_i^{\tilde{c}_{1,d_{10}}} + c_{2,d_{10}} \cdot \tau_i^{\tilde{c}_{2,d_{10}}} \cdot u^{*2} + c_{3,d_{10}} \cdot u^{*3} \tag{1}$$

$$d_{11} = c_{0,d_{11}} \cdot \tau_i + c_{1,d_{11}} \cdot \tau_i^{\tilde{c}_{1,d_{11}}} + c_{2,d_{11}} \cdot \tau_i^{\tilde{c}_{2,d_{11}}} \cdot u^{*2} \tag{2}$$

$$d_{13} = c_{0,d_{13}} \cdot \tau_i^{\tilde{c}_{0,d_{13}}} + c_{1,d_{13}} \cdot u^* \tag{3}$$

$$\gamma_{D^{(1)}} = c_{1,\gamma_{D^{(1)}}} \cdot \tau_i^{\tilde{c}_{1,\gamma_{D^{(1)}}}} \cdot u^* \tag{4}$$

    with $c_{0,d_{10}} = -0.006$, $c_{1,d_{10}} = -0.888$, $\tilde{c}_{1,d_{10}} = 0.098$, $c_{2,d_{10}} = 0.137$, $\tilde{c}_{2,d_{10}} = 0.019$, $c_{3,d_{10}} = -10.566$, $c_{0,d_{11}} = -1.656$, $c_{1,d_{11}} = -0.018$, $\tilde{c}_{1,d_{11}} = -8.853e - 05$, $c_{2,d_{11}} = -0.268$, $\tilde{c}_{2,d_{11}} = 1.671$, $c_{0,d_{13}} = -0.005$, $\tilde{c}_{0,d_{13}} = 0.012$, $c_{1,d_{13}} = 1.023$, $c_{1,\gamma_{D^{(1)}}} = 0.341$, $\tilde{c}_{1,\gamma_{D^{(1)}}} = 0.247$.

    And for the diffusion function:

$$d_{20} = c_{0,d_{20}} \cdot \tau_i^{\tilde{c}_{0,d_{20}}} + c_{1,d_{20}} \cdot \tau_i^{\tilde{c}_{1,d_{20}}} \cdot u^* + c_{2,d_{20}} \cdot \tau_i^{\tilde{c}_{2,d_{20}}} \cdot u^{*2} \tag{5}$$

$$d_{21} = c_{0,d_{21}} \cdot \tau_i^{\tilde{c}_{0,d_{21}}} + c_{1,d_{21}} \cdot \tau_i^{\tilde{c}_{1,d_{21}}} \cdot u^* \tag{6}$$

$$d_{22} = c_{0,d_{22}} \cdot \tau_i^{\tilde{c}_{0,d_{22}}} + c_{1,d_{22}} \cdot \tau_i^{\tilde{c}_{1,d_{22}}} \cdot u^* \tag{7}$$

with $c_{0,d_{20}} = 0.024$, $\tilde{c}_{0,d_{20}} = -0.0001$, $c_{1,d_{20}} = 0.0002$, $\tilde{c}_{1,d_{20}} = 1.076$, $c_{2,d_{20}} = 1.573$, $\tilde{c}_{2,d_{20}} = 1.622$, $c_{0,d_{21}} = 0.002$, $\tilde{c}_{0,d_{21}} = -0.001$, $c_{1,d_{21}} = 1.104$, $\tilde{c}_{1,d_{21}} = 1.395$, $c_{0,d_{22}} = 0.042$, $\tilde{c}_{0,d_{22}} = 0.002$, $c_{1,d_{22}} = 0.555$, $\tilde{c}_{1,d_{22}} = 0.364$.

**Page 9, figure 3: On Page 7, line 156 the authors write that the drift and diffusion terms should be polynomials of order 2 and 3. Provide fits in figure 3 to illustrate this.**

Added accordingly.

**Page 9, equation 19: Do not put 'exp' in italics.**

Changed accordingly.

**Page 10, figure 4: The black-on-black isoline notations are virtually impossible to read.**

Changed accordingly.

**Page 10, figure 4: The math in the right panel does not look convincing. Can the authors comment on this?**

As the estimation of the conditional pdfs is purely based on measurement data, sparse regions in the phase space will naturally result in poor estimations of pdfs. This is of course a major problem for our method, as it relies on those pdfs. With
the plot in the right panel we wanted to show that we now can extrapolate conditional pdfs to sparse regions via numerical solutions of the Fokker-Planck equations. Still, even though the estimated conditional pdf suffers from a low availability of data, our numerical solution appears to be a reasonable representation of the true pdf. To make our approach clear, we added in the manuscript the comment: 'Note that due the use of the FPE, the obtained pdfs are less noisy and extend to large values as seen in fig. 4'

**Page 13, line 227: Typo: replace sigma with "sigma."**

Changed accordingly.

**Page 13, line 227: Why can the coefficients D be considered slowly changing functions? Figure 7 seems to show the contrary.**

In Figure 7 we show the timeseries of the wind speed fluctuations $u*$, which are indeed fast-changing. Also the underlying conditional pdfs $p(u^*|u_1;...;u_N)$, shown to the right, are fast-changing, due to different values of $u_1$, ..., $u_N$. It is a central point of our multipoint approach that the underlying Fokker-Planck equation, defined by its coefficients $D^{(i)}$, is not changing for this interval, or, respectively, is slowly changing due to the non-stationary nature of windspeeds. This slow change takes place on large time scales for which the mean values like $\overline{U}$ are defined. In the same sense the joint pdfs $p(u^*;u_1;...;u_N)$ are changing only slowly, while the fast-fluctuating conditional pdfs $p(u^*|u_1;...;u_N)$ are generated by the fast-fluctuating arguments in the joint pdf. The basic difference is based on the aspect one is looking at. The signal is fast changing, but the underlying equation is fixed. Fast changes are generated by fast-fluctuating arguments of a slow-changing function.

**Page 14, figure 8: Consider splitting the top panel into two. Through the overlapping curves, a lot of detailed information is getting lost.**

**Page 15, line 248: RANS should be spelled out.**

**Page 15, line 249: 'Great' is an odd choice of word.**

**Page 15, line 255: Consider rephrasing the sentence. Not all multipoint-based models automatically capture small-scale intermittency.**

**Page 15, line 265: Do not use capitals in the reference. Page 15, line 267: Capital 'A' in acknowledgments.**

All changed accordingly.

**Page 15, line 269: Is the first name of the person Andeé?**

No, thank you for pointing this out.

**Page 16 References: A lot of the references are inconsistent when it comes to providing doi, placement of first name letters and abbreviation dots.**

Changed accordingly.

**References**

Boettcher, F., Renner, C., Waldl, H.-P., and Peinke, J.: On the statistics of wind gusts, Boundary-Layer Meteorology, 108, 163–173, https://doi.org/10.1023/A:1023009722736, 2003.

Morales, A., Waechter, M., and Peinke, J.: Characterization of wind turbulence by higher-order statistics, Wind Energy, 15, 391–406, https://doi.org/10.1002/we.478, 2012.

Peinke, J., Tabar, M. R., and Wächter, M.: The Fokker–Planck Approach to Complex Spatiotemporal Disordered Systems, Annual Review of Condensed Matter Physics, 10, 107–132, https://doi.org/10.1146/annurev-conmatphys-033117-054252, 2019.

---

## Author Comment (AC2) · 2 Jun 2020

**Reply to anonymous Referee No.2: Multipoint Reconstruction of Wind Speeds**

Christian Behnken, Matthias Wächter, and Joachim Peinke

Institute of Physics/ForWind, University of Oldenburg.

**Correspondence:** Peinke (peinke@uni-oldenburg.de)

*Copyright statement.* TEXT

**NOTE: All figure and equation numbers refer to the original submitted manuscript and may differ from the ones in the revised version.**

**L13. I would argue that hydro is more represented in decarbonized energy sources than wind and solar; at least in some parts of the world. It should be included in this list.**

We added it to the list in the revised manuscript.

**L15. Why wind(solar) is capital in the manuscript?**

**L25. The citation style is incorrect. Please revise accordingly.**

**L28. Should be "...known to be..."**

**L33. There should be a space between 10 and min. Please apply the same correction everywhere else (number and unit separation). Also, the citation style is incorrect. Please revise this issue everywhere in the manuscript.**

All changed accordingly.

**The exact definition of intermittency (for the context of this study) should be provided in the Introduction. The authors talk a lot about intermittency, but the exact definition is not provided.**

We added following definition in the introduction:

"Within this context the term intermittency is used in the spirit of Kolmogorov 62 to describe the characteristic heavy-tailed shape of pdfs often found at small scales in time series of turbulent systems (Frisch, 2004)."

25    **L46. Remove one "and" at the end of this line.**

**L60. It should be specified that t is the time.**

**L61–62. Please revise the sentence for proper English.**

All changed accordingly.

30

**L60 and L65. Please clarify the difference between u(t) and U(t).**

We clarified it in the revised manuscript in the beginning of the method section:

"With $U(t)$ we refer to the resulting wind speed from the horizontal components. The quasi-stationary wind speed $u(t)$ is then

35    obtianed from $U(t)$ by respectively normalizing it with the mean $\overline{U}$ and standard deviation $\sigma_U$ within blocks of 1 min length."

**The abbreviation pdf is sometimes italicized and sometimes not. Please be consistent.**

**L110. There should be a comma after the Pawula theorem. Also, please provide a reference for this claim on L110 and L111.**

40

All changed accordingly.

The reference for the Pawula theorem is Risken 1996. Regarding our claim in L110 und L111, we added following plot to

the manuscript: "As one can see in fig. (1), the fourth Kramery-Moyal coefficient is slightly larger than zero, but negligible

45    compared to the magnitude of the diffusion function $D^{(2)}$"

[Figure]

**Figure 1.** Exemplary estimations of the second and fourth Kramers-Moyal coefficient $D^{(2)}$ and $D^{(4)}$ for $\tau = 65\ s$

**L155–156. Why the order of the polynomial of 3 and 2. Is this the lowest polynomial order that properly fits the data?**

Indeed third and second order polynomials for the drift $D^{(1)}$ and diffusion $D^{(2)}$ functions are the polynomials of lowest that are properly fitting the data. Emperical studies ((Renner et al., 2001), (Reinke et al., 2018)) suggests that these polynomials
50 are well suited to problems in fluid mechanics. Choosing higher order polynomals is possible as well, but the parametrization might be suffering from overfitting then. Furthermore, up to now we did not see any fundamental changes in the results using higher order polynomials – see a rigorous approach to support these findings by the use of the integral fluctuation theorem for ideal turbulent data (Reinke et al., 2018).

55 **Figure 2. The two labels in the legend are identical, but the different notation is used in the figure caption. Please correct this before this figure can be reviewed properly.**
**L167. Please correct the English.**
**All figures. Please add (a), (b), (c), etc. labels for subplots.**
**L179. I belie that "an" should be "a".**
60 **Equation 19. The function exp should not be italicized. The same holds for any other function in the manuscript.**
**L223. The word min should not be italicized.**

All changed accordingly.

65 **MAJOR COMMENT: L226. In non-stationary wind speed records, the fluctuations are dependent on wind speed. Reading this section (and this particular line), this reviewer concludes that the presented methodology does not account for this relationship. For instance, in the case of non-stationary thunderstorm winds, Chen and Letchford (2004) (doi: 10.1016/j.engstruct.2003.12.009) modulated the fluctuations based on the moving-mean wind speed. A similar approach was used by Chay et al. (2004) (doi: 10.1016/j.engstruct.2005.07.007). This has been shown on the example of full-scale**
70 **data of thunderstorm winds in Burlando et al. (2017) (doi:10.1175/MWR-D-17-0018.1) and Zhang et al. (2018) (doi: 10.1016/j.probengmech.2017.06.003). Notice that in these papers the moving-mean turbulence intensity in the transient (thunderstorm) wind record is not changing in time. This confirms that the fluctuations increase as the mean wind speed increases. Please clarify this issue because it is particularly important for transient wind speed records. This change (previous comment) would perhaps correct for the discrepancies between the measurements and the reconstruction in**
75 **Figure 8 (pdfs).**

This comment of the referee addresses several points to which we want to answer:

**Comment on Chen and Letchford (2004) (doi: 10.1016/j.engstruct.2003.12.009)**: In this paper special wind situations of
80 thunderstorm downbursts are grasped by a deterministic–stochastic hybrid model. The fluctuation is modeled as a uniformly

modulated evolutionary vector stochastic process. In our paper we focus on this stochastic part and not on the larger scale deterministic part as (Chen and Letchford, 2004). In contrast to (Chen and Letchford, 2004) we do not model the fluctuations by a stochastic process in time, but we show that a new class of a sstochastic process in scale can be used. Common stochastic processes are Markovian in time and thus are not able to grasp general aspects of multi-point statistics. Turbulent wind signals are in general not Markovian in time, but it is the novelty that we show in our paper that these turbulent wind fluctuations are Markovian with respect to a special scale process (see fig. 2), which enables us to set up a stochastic process in scale. This scale process is more complicated, but statistically more complete.

**Cases of rapidly changing wind conditions like thunderstorm events or other transient wind speed changes are not in the focus of our work**. We aim at modelling the quasi-stationary wind speed fluctuations $u^*(t)$, which are obtained by a blockwise normalization (of 1 min length) with respect to the mean and standard derivation of wind speeds $U(t)$. This way we decouple the fluctuations from the magnitude of the mean flow. As stated later on, a rescaling of the fluctuations is achieved when we transform the modelled fluctions $u^*(t)$ back to real wind speeds $U^*$, by multiplying it with the standard derivation: $U^* = (\sigma_U \cdot u^*) + U$.

If and how our approach may be adapted to situations like thunderstorms is out of the scope of our paper, may be just a shortening of our decompositioning in 1 min - blocks is already helpful. We would agree to add this point as a fotenote in our paper or add it to the discussion.

**Concerning the discrepancy of Fig 8**:

In our data there was no thunderstorm like behavior. The discrepancy is mainly statistical nature. We have two comments to the discrepancies in the plots of fig. 8:

a) Discrepancy in the timeseries: The mean wind speeds $\overline{U}(t)$ were generated by simple stochastic model and thus there will be deviations from the corresponding mean wind speeds from the measurements to the very same timestamp. If we would have used the historic mean wind speeds, there would be only minor deviations.

b) Discrepancy in the increment pdfs: The main deviations in the incrementd pdfs are found at the tails of the pdf on larger scales $\tau_i$ (note also the logarithmic y-scale). As these are probabilities our model is virtually completely correct for all scales, especially for small scales $\tau_i$.

**MAJOR COMMENT: Related to my previous comment, non-stationary velocity records are often non-Gaussian too. Can you please clarify how is this accounted for in your methodology?**

This is correct. The central point for our approach here is the validity of the Markov property. If this is fulfilled, the other parts, like the shape of the probability distributions, are mathematically rigorous consequences. As mentioned above, in a careful investigation one may find wind conidiations for which our approach is not valid. The importance of such cases are out of

the scope of our work presented here.

115

**MAJOR COMMENT: The purpose of this methodology is to generate fluctuating wind records. This topic addressed in the seminal paper by Shinozuka (1972) (doi: https://doi.org/10.1016/0045-7949(72)90043-0). Without going into mathematical rigor in this review, the basis of his method is to generate random numbers (through Monte Carlo) that follow the prescribed power spectral density of wind fluctuation (e.g., Kamal spectra, Davenport spectra, von Karman spectra,**

120 **Mann spectra, etc.). This method is later implemented in some of the studies provided in my comment 20 and references therein. So, my question is how the method proposed in your study extends beyond this well-established methodology of generating wind fluctuations? What are the benefits of using the presented method in your study?**

Methods relying on a prescribed power spectral density (PSD) to generate time series of wind speed fluctuations do have

125 the benefit of being computationally fast and applicable without posing much requirements on the data. Nevertheless such methods only provide time series of a predefined length as shown for the amplitude-modulation sheme in (Chen and Letchford, 2004). The benefit of our method is that a time series can be continued in-situ for an arbitrary amount of iterations. Due to the stochastic nature of our algorithm an ensemble of possible scenarios for the evolution of the wind speed fluctuations, starting from a specific situation, can be assembled.

130 Furthermore, the main point of our paper is that we are mathematically much more general as Kamal spectra, Davenport spectra, von Karman spectra and Mann spectra, which are all low order two point (two time quantities, (see (Peinke et al., 2019)). Thus intermittency (higher order two point quantity) like fig. 8 and more complex multipoint structures (like gusts, see fig. 7) are now grasped by our approach. Our paper will open a new way to investigate such data (see (Fuchs et al., 2020) and (Hadjihosseini et al., 2016)). Note that the knowledge of the Fokker-Planck equation describes in a very compact way all the

135 changes in statistics of two point (time) quantities as shown in fig 8.

**MAJOR COMMENT: Can the authors plot spectra of the two velocity time series in Figure 5? Please also include the reference –5/3 slope for benchmarking.**

140 We see a good agreement between the spectra from the mesaurements and the reconstruction with the -5/3 spectra within the internal subrange between $f > 0.1\,Hz$ and $f < 1\,Hz$ (see fig. 2). The flattening of the spectra observed at low frequencies ($f < 0.1\,Hz$), was also noted by (Morales et al., 2012) for wind speeds in a similar range. But as this observation is not of interest for our work, we do not discuss further details here.

Furthermore we would like to stress that the spectrum from the reconstructed time series matches the one from the measurement

145 very well, disregarding the deviations at high frequencies, where we are in the range of measurement noise of the ultrasonic anemometer. This shows that our method is able to capture two-point statistics like the power spectral density, but we would like to note, that we are going beyond, as the generated wind speed fluctuations $p(u^*, t^* | u_1, t^* - \tau_1; ...; u_N, t^* - \tau_N)$ are based

on N-point statistics.

[Figure]

**Figure 2.** power spectral density of measured and reconstructed wind speed fluctuation

**MAJOR COMMENT: Going back to L36 in the Introduction. The authors correctly talk about the spatial dependency of fluctuations and coherence. How is the current model generating fluctuations in space? The presented results are for a point measurement, but the implementation for wind energy (i.e., wind turbine) analysis requires the spatially dependent profile. How a coherence function can be implemented in the method?**

The main scope of our paper is to present a new method to generate realistic time series of wind speed fluctuations. An extension to higher dimensions, enabling one to generate wind fields in time and space is of course desirable, but the authors would consider this to be the next step, as this will not be straightforward to do. We have three further remarks on our approach:

a) The one-point time signal corresponds in the common approach of Taylor's frozen turbulence hypothesis to spatial structures in the flow direction, regardless some necessary correction to Taylor's hypothesis.

b) Our approach is statistically complete for one direction (in the sense of grasping any n-point statistics), thus the question to extend this to the full three-dimensional space would run into a statistical solution of the turbulence problem, which is still our dream to pave the way.

c) The knowledge of a one point-time series already provides a better prediction of loads and power outputs as shown by (Wächter et al., 2010)

**L238. The phrase "a fairly nice match" is not scientific. Please be specific.**

We reformulated our comparison in the revised manuscript.

**170** **How computationally efficient is your method? How much computational time is required to generate a fluctuation time series of different lengths? Can you please comment on this?**

One step in scripting languages like Python or R takes on average about 0.0005 s at generating a time series of $10^4$ length on an ordinary PC. Utilizing langues like C/C++ or Fortran the computation should be boosted at least with a factor 10-100. **175** So the decline in processor power upon generating very large time series will not be of much impact for the practicability of our method. It is also not the computational efficiency which we emphasize here, but the new quality (mutli-point statistics) we give access to by this apporach.

**L252–L253. Not necessarily until the method accounts for the spatially coherent fluctuations.**

**180** We agree that this is an important aspect. Thus we suggest to add a footnote in our article to clarify this point:
"Note, here we do not include the aspect of spatial coherence. To affect a big WEC such temporal fluctuations must have a sufficient large spatial structure."

**References. Some citations include article titles while the others do not. In addition, some journal names are abbre-** **185** **viated whereas the others are not. Please be consistent.**

Changed accordingly.

**Title: what exactly the authors mean by "multipoint?" This reviewer assumes this word signifies the time dependency** **190** **of the methodology. If yes, isn't this redundant because fluctuations have to be time dependent?**

We agree that the term "multipoint" needs to be specified. We will add following explanation in the introduction:
"While commonly applied methods, like spectral analysis and two-point correlations, limit themselves to two-point statistics, here we extend the methodology to more than two points in time. We obtain generalized correlations between multiple points **195** in time, in terms of probabiliy density functions (pdfs) for the occurrence of a whole sequence of wind speeds. Those pdfs we denote multipoint pdfs, and they constitute the basic concept of our approach."

**References**

Chen, L. and Letchford, C. W.: A deterministic–stochastic hybrid model of downbursts and its impact on a cantilevered structure, Engineering Structures, 26, 619–629, https://doi.org/10.1016/j.engstruct.2003.12.009, 2004.

Frisch, U.: Turbulence: The Legacy of A. N. Kolmogorov, vol. 1, Cambridge Univ. Press, 2004.

Fuchs, A., Queirós, S. M. D., Lind, P. G., Girard, A., Bouchet, F., Wächter, M., and Peinke, J.: Small scale structures of turbulence in terms of entropy and fluctuation theorems, Phys. Rev. Fluids, 5, 034 602, https://doi.org/10.1103/PhysRevFluids.5.034602, 2020.

Hadjihosseini, A., Wächter, M., P.Hoffmann, N., and Peinke, J.: Capturing rogue waves by multi-point statistics, New J. Phys, 18, 013 017, https://doi.org/10.1088/1367-2630/18/1/013017, 2016.

Morales, A., Waechter, M., and Peinke, J.: Characterization of wind turbulence by higher-order statistics, Wind Energy, 15, 391–406, https://doi.org/10.1002/we.478, 2012.

Peinke, J., Tabar, M. R., and Wächter, M.: The Fokker–Planck Approach to Complex Spatiotemporal Disordered Systems, Annual Review of Condensed Matter Physics, 10, 107–132, https://doi.org/10.1146/annurev-conmatphys-033117-054252, 2019.

Reinke, N., Fuchs, A., Nickelsen, D., and Peinke, J.: On universal features of the turbulent cascade in terms of non-equilibrium thermodynamics, J. Fluid Mech, 848, 117–153, https://doi.org/10.1017/jfm.2018.360, 2018.

Renner, C., Friedrich, R., and Peinke, J.: Experimental indications for Markov properties of small-scale turbulence, J. Fluid Mech, 433, 383–409, https://doi.org/10.1017/S0022112001003597, 2001.

Wächter, M., Mücke, T., and Peinke, J.: Influence of vertical shear and turbulence intensity on Langevin power curves, DEWEK Bremen, 2010.